# Directionally-Enhanced Binary Multi-Objective Particle Swarm Optimisation for Load Balancing in Software Defined Networks

**DOI:** 10.3390/s21103356

**Published:** 2021-05-12

**Authors:** Mustafa Hasan Albowarab, Nurul Azma Zakaria, Zaheera Zainal Abidin

**Affiliations:** Fakulti Teknologi Maklumat Dan Komunikasi, Universiti Teknikal Malaysia Melaka (UTeM), Hang Tuah Jaya, Durian Tunggal, Melaka 76100, Malaysia; p031710041@student.utem.edu.my (M.H.A.); zaheera@utem.edu.my (Z.Z.A.)

**Keywords:** SDN, load balancing, multi objective particle swarm optimisation, crowding distance, angle searching, execution time, energy consumption, less execution cost

## Abstract

Various aspects of task execution load balancing of Internet of Things (IoTs) networks can be optimised using intelligent algorithms provided by software-defined networking (SDN). These load balancing aspects include makespan, energy consumption, and execution cost. While past studies have evaluated load balancing from one or two aspects, none has explored the possibility of simultaneously optimising all aspects, namely, reliability, energy, cost, and execution time. For the purposes of load balancing, implementing multi-objective optimisation (MOO) based on meta-heuristic searching algorithms requires assurances that the solution space will be thoroughly explored. Optimising load balancing provides not only decision makers with optimised solutions but a rich set of candidate solutions to choose from. Therefore, the purposes of this study were (1) to propose a joint mathematical formulation to solve load balancing challenges in cloud computing and (2) to propose two multi-objective particle swarm optimisation (MP) models; distance angle multi-objective particle swarm optimization (DAMP) and angle multi-objective particle swarm optimization (AMP). Unlike existing models that only use crowding distance as a criterion for solution selection, our MP models probabilistically combine both crowding distance and crowding angle. More specifically, we only selected solutions that had more than a 0.5 probability of higher crowding distance and higher angular distribution. In addition, binary variants of the approaches were generated based on transfer function, and they were denoted by binary DAMP (BDAMP) and binary AMP (BAMP). After using MOO mathematical functions to compare our models, BDAMP and BAMP, with state of the standard models, BMP, BDMP and BPSO, they were tested using the proposed load balancing model. Both tests proved that our DAMP and AMP models were far superior to the state of the art standard models, MP, crowding distance multi-objective particle swarm optimisation (DMP), and PSO. Therefore, this study enables the incorporation of meta-heuristic in the management layer of cloud networks.

## 1. Introduction

Recent years have seen a rapid boom in the development of many new technologies such as Internet of Things (IoTs) and cloud systems. The emergence of cloud computing and data storage centres has led researchers to focus on optimising functionality and service. The implementation of these optimisations became easier and flexible with the development of software defined networking (SDN). However, the centralized control architecture of SDN generates concerns about reliability, scalability, fault tolerance, and interoperability [1]. Hence, assuring efficient management of SDN based computation is essential for the successfulness of the system. Computing, a common type of service type, can be defined as coordinating the execution of big processes or tasks on a network to meet a set of objectives or goals such as maximum reliability, minimum execution time, and minimum rental cost. Simultaneously, meeting all these is not possible due to implicit conflicts. Hence, the incorporation of multi-objective optimisation (MOO) is needed to define an optimisation point from a set of non-dominated points. Meta-heuristic based MOO optimisation is a suitable algorithm candidate that can be adapted for this purpose. Meta-heuristic-based SDN load balancing has been suggested as a new and emerging research topic [2].

Meta-heuristic searching algorithms are a family of algorithms that follow the same concept of searching within numerous or infinite sets of candidate solutions to find the best non-dominated solutions, the optimal solutions, to a problem with respect to one or multi-objective functions. Most meta-heuristic searching algorithms are associated with certain semaphores such as black holes [3], simulated annealing [4], bee colonies [5], and genetics [6]. They only differ in the nature of their search and their ability to avoid local minima while completing the search in a minimum amount of time. However, meta-heuristic searching was originally developed as a single-objective optimisation. Considering the multi-objective nature of many real-world problems, researchers have endeavoured to convert many single-objective optimisations to MOO such as the multi-objective non-dominated sorting genetic algorithm II (NSGA-II) [7] and NSGA-III [8]. For particle swarm optimisation (PSO), multi-objective particle swarm optimisation (MOPSO) [9], many other variants [10], and [11] were developed. Various tools and frameworks were also provided to compare MOO algorithms and develop their features.

Analysing multi-objective performance requires relying on non-dominated sorting to evaluate solutions and travel change within the search space of a meta-heuristic particle swarm. However, non-dominated sorting is not the only means in evaluating the solution in the swarm. Other aspects of performance, such as the diversity of the solutions and their spread in the decision space, warrant identification. Hence, other criteria are used by researchers such as crowding distance in NSGA-II [12] and directional distribution of solutions in angle quantization [13]. However, an integrated criterion capable of combining both angle and direction has yet to be used effectively in a MOPSO variant model.

Computational load balancing of tasks, or tasks allocation, on a set of computing nodes is a multi-objective combinatorial optimisation problem (MOCOP) where the goal is to minimise the execution time and the cost of renting the necessary nodes. However, the problem is regarded as a non-deterministic polynomial time NP-hard optimization problem which motivates researchers to use meta-heuristic searching to solve it. Furthermore, the problem is a discrete searching problem. Hence, solving this type of optimisation problem requires searching in discrete spaces with optimised multi-objective functions while being mindful of the execution time and the node renting cost.

There are two categories in the context of load balancing using SDN technology—computational load balancing and traffic load balancing. The objective of this study was to present new joint mathematical formulation for load balancing problems from the perspectives of computational load balancing, namely, execution time, energy consumption, and cost. The study also aimed to develop a novel MP variant model with enhanced exploration, crowding distance, and angle quantization. Two variant models, continuous and discrete, were developed and compared with existing MP and crowding-distance-based multi-objective particle swarm optimisation (C-MOPSO).

This study presents a set of contributions. We summarize them as follows.

(1) It proposes a joint mathematical formula that addresses three load balancing problems: time, energy, and cost.

(2) It proposes a MOO-based PSO algorithm which not only takes into account the directions of the solutions in the space but also provides a novel solution selection algorithm by combining crowding distance and angle, distance angle multi-objective particle swarm optimisation (DAMP), or using the angle criterion at one time, angle multi-objective particle swarm optimisation (AMP).

(3) It evaluates the developed MOO-based PSO algorithm using mathematical functions and it compares with benchmarking algorithm from MOO-PSO literature. In addition, binary variants of the approaches were generated based on transfer function and they are denoted by B-(Method Name), e.g., BAMP for AMP.

The literature review is presented in Section 2, and the methodology is detailed in Section 3, while the evaluation and the results are explained in Section 4. The conclusion and the future considerations are expressed in Section 5.

## 2. Literature Review

The problem of load balancing in SDN networks has become an active research topic in the recent years. In the work of [14], the architecture of SDN layers considers load balancer as one block of SDN application tier. In the survey of [15], the authors provided a taxonomy of load balancing in SDN with discussing the various objectives such as response time, resources optimization throughout, and bottlenecks. Their taxonomy has classified load balancing in control plane and data plane. The former was divided into hierarchal and virtualization controller, while the latter was divided into server and link.

There are many research articles exploring the use of meta-heuristic searching in SDN-based network applications, for instance, modified genetic searching for SDN placement in networks [16] and chaotic salp swarm algorithm (CSSA) optimisation to obtain the optimal number of SDN in networks [17], while [18] proposed a resource selection MOO genetic algorithm using SDN network. There are two categories in the context of load balancing using SDN technology—computational load balancing and traffic load balancing. In the work of [19], an approach for enabling real-time traffic matrix for the traffic measurement system in SDN was proposed. Their design includes fixed and elastic schemas in order to achieve overhead reduction without compromising on accuracy. Hence, it falls under the category of multi-objective SDN network traffic measurement. In the work of [20], a flow-aware elephant flow detection applied to SDN was proposed in order to enable sharing the elephant flow classification tasks between the controller and the switches, which is a type of traffic load balancing.

Many studies have used meta-heuristics for traffic load balancing, such as a study by [21], where genetic algorithm was integrated with ant colony optimisation for traffic load balancing, and a study by [22] that proposed genetic optimisation for traffic loading using SDN. In this study, the researchers used a load balancing algorithm to identify the shortest path, which requires the least number of operations, by looking for the lowest capacity among the switches. This was determined by the load balancer using information provided by the controller. Three pieces of information were used for this objective: the path cost, the switch capacity, and the operational cost. The algorithm was also designed to have a mutation operation that used a path and a link to create a new path [22]. While that study only merely used a single optimisation, other studies have adopted MOO algorithms for similar purposes [16]. Computational load balancing aims to optimise task execution from various perspectives such as execution, reliability, and cost. In a study by [23], a framework for load balancing using two meta-heuristic optimisation methods focused on makespan and cost metrics. However, their framework failed to evaluate exploratory searches. Another work by [24] used MOO for SDN-based load balancing where quality of service (QoS) was considered a constraint, while energy saving and load balancing were optimised using MOPSO. This method also failed to evaluate the exploration or the diversity of the solutions. Load balancing has been applied in advanced simulations such as molecular dynamics in a study by [25], where the optimisation was based on heterogeneous supercomputers, which made the optimisation more difficult or a non-deterministic polynomial time NP-hard problem and combined genetic optimisation and PSO.

Numerous recent studies explore developing effective MOPSO with a focus on how to best supply searching via a combination of methods to enhance and obtain more optimal solutions. In a study by [26], four strategies—multi-population, dynamic clustering, solution life, and probability lottery—were used in conjunction with their MOPSO variant model. The study concluded that their MOPSO variant model was superior since it included more than one strategy while searching. However, it failed to carefully evaluate other strategies such as set coverage, hyper-volume, and delta measure, thereby rendering their algorithm inadequate. Another aspect that is currently under consideration is the computation cost of an algorithm when multi-populations are added.

In another study [27] using MOPSO, the moving strategy of particles toward local and global was embedded in the cross-over operation, which cannot be considered a real improvement in the search itself. The MOPSO variant models developed by the studies mentioned in this literature review used various concepts to ensure the discovery of an adequate set of non-dominated solutions. In a study by [28], MOPSO was used with crowding distance to perform clustering. Considering that multi-objective optimisation is evaluated using wide set of indicators [29], various approaches were developed based on the concept of using one of the indicators of diversity to select solutions from one iteration to another. In another study, R2 indicator contribution was used to select particles instead of crowding distance value, which is capable of achieving higher diversity in the search [30]. R2 indicator contribution value was used also by another MOO-based PSO method study in order to scalarise the solutions in the archive of non-dominated solutions [31]. Another study used unary epsilon indicator and Pareto dominance [32] in addition to direction-based reference points similar to a study by [33].

A study by [34] used angle-based searching in MOO-based PSO, which considered selecting solutions from the low density angle region and deleting the extra particles from the high density angle region. While the method was based on adaptive angle division, distance-based crowding was not included in the search, which affected one aspect of the diversity of the discovered solutions. Other researchers have developed MOPSO that incorporated crowding distance [35]. The study modified the velocity formula by including the sharing-learning factor. The sharing factor was added as a third term in the velocity equation to move the particle not only to the direction of personal and global beat particles but also to the average of all other particles in the swarm, which added more diversity to the swarm. The approach also added Gaussian mutation to the particles to achieve higher exploration. Furthermore, the study proposed an update of best global using greedy strategy with respect to each particle’s changing position. This method, however, ignored the mobility direction in the particles, which is regarded as another factor in the diversity of exploration. A study by [36] evaluated an aspect that is usually ignored, leaders evaluation in traditional PSO. They went on to propose a new concept that a good leader takes feedback from his/her followers and modifies their decisions accordingly. The study went on to identify various cases of follower-based improvements in the swarm according to the change of the fitness values. Based on that, the velocity of the leader which expressed the changes of the movement speed and direction toward the leaders was changed. However, this method was only applied in single-objective optimisation. Another MOPSO variant model, which incorporated a new concept to achieve diversity and exploration, was proposed by [37]. The researchers divided the space into sets of hyper-boxes and tracked the number of solutions in each hyper-box, finally considering only the solutions at the boundary of each hyper-box. Some methods have developed new criteria for selecting leaders from the repository of non-dominated solutions. For example, a study by [38] developed an improved MOO variant model and provided an algorithm for selecting leaders from sets of non-dominated solutions using a geometrical approach. The approach selected points that had the least distance from the line fitted model for the set of non-dominated solutions. However, the problem with this approach was its invalid assumption of the straight-line approximation of the set of non-dominated solutions occurring in most cases.

Other studies have incorporated clustering in solution selection from one iteration to another, where the set of non-dominated solutions were decomposed to clusters and solutions belonging to different clusters were selected to achieve diversity in the solutions. For instance, [39] used Euclidean distance for clustering. This can be criticised for the fact that an implicit sphere assumptions of Pareto parts geometric model was made, which is not valid in many types of optimisation surfaces. Many other methods have opted to convert MOO to mono-objective optimisations via decomposition and use it along with dominance to select solutions. This was done in a study by [40] where penalty-based boundary intersection (PBI) was used with dominance to create a hybrid strategy. In our opinion, researchers can fall in the non-convexity trap by applying mono-objective mapping. In the same vein, a study by [41] estimated solution domination using cosine transformation and reference vector association and also used simplified leader-oriented mobility equation to counter the slow convergence and simplify the calculation. The approach presented elite velocity-based selections and a twofold leader definition. However, the cosine distance and the reference vector association lacked accurate estimation of solution diversity in the search space. Studies that have developed MOO-based PSO by exploiting existing information and communication theory concepts, namely entropy and its usefulness in probing the convergence of algorithms based on the entropy behaviour, also warrant mention. This method was first proposed by [42] where a simulation was used to prove the association between the change of entropy in the particles of the Pareto front and the convergence of the algorithm. This method was further developed by [43] where, using particle entropy, the particles were mapped on a parallel-cell coordinate system and a feedback information system, and the difference of entropy was used to change the parameters of the algorithm. Although entropy can be a useful metric indicator of the diversity of the solutions, it still lacks actual geometric and directional description of the particles in the space.

While numerous studies have evaluated load balancing using meta-heuristic, none have evaluated it from the perspective of non-dominated solutions. Furthermore, the exploration of the solution space has not received adequate attention since it is critical in providing the decision maker with sets of choices. Although MOPSO was among the evaluated methods, the improvements done to it ignored searching direction, which plays an important role in the diversity of the offered solutions. A summary of the models and the objectives listed in this literature review are presented in Table 1. None of these studies explored incorporating all five objectives in one model. Therefore, we present the findings of our MP-based load balancing model in subsequent sections.

## 3. Methodology

The symbols used throughout this study are explained in Table 2. The methodology starts with presenting the network model in Section 3.1. Next, the task model is provided in Section 3.2. Afterwards, the energy model, the time execution metric, and the renting cost metric are provided in Section 3.3, Section 3.4 and Section 3.5, respectively. Next, the optimization objective functions are given in Section 3.6. Next, we present the transfer function model for dealing with the binary space in Section 3.7. The crowding distance is presented in Section 3.8, and the developed DAMP algorithm is provided in Section 3.9. Next, a big O notation is given in Section 3.10. Lastly, a separated section is dedicated for the evaluation analysis.

### 3.1. Network Model

The network was represented by an undirected graph $G\left(N,R\right)$, where the networks were $N=\left\{N_{j}:j=1,2,\ldots n\right\}$
$R=\left\{e_{ij},i,j=1,2..r\;and\;i\neq j\right\}.$ The edge between two nodes (i,j) had a weight that represents the distance between the two nodes ($d_{ij}$). When the i node was not connected to the j node, the distance between the two was infinity. When unconnected, the nodes were represented as $N_{5}$ and $N_{6}$.

Each node was described by variables to determine its computational and power specifications. In order to describe node $n_{j}$, we used the tuple $\left(v_{j},e_{j},e_{init}\right)$ where $v_{j}$ denoted computation power of the node, which was measured as instruction per second (IPS), $e_{j}$ denoted average energy consumption as measured by Joules per second (J/sec), and $e_{init}$ denoted the initial energy. Each node was also described by two constant variables, $P_{0}$ and $L_{0}$. $P_{0}$ represented the maximum computational load, while $L_{0}$ represented the maximum communication load.

### 3.2. Task Model

The task model $G\left(M,E_{M}\right)$ was provided by directed acyclic graph (DAG) where we assumed that we had m tasks $M=\;\left\{M_{i}:i=1,2,\ldots m\right\}$, and each task was described by computation or, $P=\;\left\{P_{i}:i=1,2,\ldots m\right\}$, which indicated the number of instructions (NI) and the communication or, $L=\;\left\{L_{i}:i=1,2,\ldots m\right\}$, measured in bytes. Since the nodes that executed the tasks had $P_{0}$ and $L_{0}$, Equation (1) was used to determine the number of nodes required to execute any i task as K.
(1)$$K=\max\left(\lceil \frac{P_{i}}{P_{0}}\rceil ,\lceil \frac{L_{i}}{L_{0}}\rceil ,K_{0}\right)$$
where $\;K_{0}$ denotes the minimum number of nodes required to execute one task.

### 3.3. Energy Model

The energy consumption model was a combination of two parts; the first part, $E_{comp}$, denoted energy consumption based on the execution of the instructions of the $P_{i}$ task and was determined using Equation (2).
(2)$$E_{comp}=\frac{P_{i}}{v_{j}}\times e_{j}$$

The second part, $E_{comm}$, was a combination of two other parts and was expressed in Equation (3).
(3)$$E_{comm}=E_{Tx}\left(k,d\right)+E_{Rx}\left(k\right)$$

It assumed that the radio emitted $E_{elec}$ [$\frac{nJ}{bit}$] to power the transmitter or the receiver circuit and was expressed in Equation (4).
(4)$$E_{Tx}\left(k,d\right)=E_{Tx-elec}\left(k\right)+E_{Tx-amp}\left(k,d\right)=E_{elec}\times k+\epsilon_{amp}\times k\times d^{2}$$

In order to receive, the energy was expressed in Equation (5) as follows:(5)$$E_{Rx}\left(k\right)=E_{Rx-elec}\left(k\right)=E_{elec}\times k$$
where:$E_{Tx}\left(k,d\right)$ was the energy consumption when transmitting k  bits over a distance d,$E_{Rx}\left(k\right)$ was the energy consumption when receiving k  bits,k was the number of transmitted bits and derived from L in the task model,d was the distance between the two nodes and derived from the network model,$E_{elec}=50\;\left[\frac{nJ}{bit}\right]$ was the constant required to power the transmitter or the receiver circuit, and$\epsilon_{amp}$ was the coefficient related to the transmitter amplifier and equalled 100 $\left[\frac{pJ}{bit}/m^{2}\right]$
(6)$$E_{ij}=E_{comm}+E_{comp}=E_{Tx}\left(k,d\right)\;+\;E_{Rx}\left(k\right)\;+\;E_{comp}$$

Using $L_{i}$ which was communication load of task i and substituting Equations (2), (4) and (5) in (6) provided (7)
(7)$$E_{ij}=E_{elec}\times L_{i}+\epsilon_{amp}\times L_{i}\times d^{2}+E_{elec}\times L_{i}+\frac{P_{i}}{v_{j}}\times e_{j}$$

### 3.4. Time Execution Metric

Equation (8) expresses how long each node ($n_{i}$), at a velocity of $v_{i}$, spent executing task j two times. Its computation and communication are given as follows:(8)$$t_{ij}=t_{com}^{ij}+t_{comp}^{ij}+t_{queue}^{ij}=\frac{L_{j}}{B}+\frac{p_{j}}{v_{i}}+t_{queue}^{ij}$$

Equation (9) was used to determine the makespan
(9)$$T=\overset{}{\underset{i,j}{\sum }}t_{ij}$$

### 3.5. Renting Cost Metric

By assuming that each node in the network had a renting rate of $r_{i}$, and assuming that the $n_{i}$ node needed to operate for $t_{ij}$ in order to execute task j, then the total renting cost of the node was Equation (10)
(10)$$RC_{ij}=r_{i}t_{ij}$$

However, to minimize the renting cost of all nodes to execute all tasks, we used the Equation (11)
(11)$$RC=\overset{}{\underset{i,j}{\sum }}RC_{ij}$$

### 3.6. Optimisation Objective Functions

The solution was optimised by assigning specific tasks to specific nodes as defined by the matrix $X=\left\{x_{ij}\right\}\in {\left\{0,1\right\}}^{n\times m}$. Therefore, the problem was a binary optimisation problem. The objective function was described by the five following Equations (12)–(16)
(12)$$f_{1}=\overset{}{\underset{i,j}{\sum }}E_{ij}$$
(13)$$f_{2}=\overset{n}{\underset{i=1}{\sum }}\frac{{\left(E_{i}-\bar{E}\right)}^{2}}{n-1}$$
(14)$$f_{3}=T$$
(15)$$f_{4}=RC$$
s.t
(16)$$k=\max\left(\frac{P_{i}}{P_{0}},\frac{L_{i}}{L_{0}},k_{0}\right)$$
and the connectivity limitation of the dependent tasks.

### 3.7. Transfer Function Model

Transfer function was used to convert particle swarm searching to binary to solve the described problem. Assuming that the i particle had a velocity of $v_{i,d}^{t}$ for the dimension d and the iteration t, the corresponding particle bit changed its value with a probability of $TF\left(v_{i,d}^{t}\right)=\frac{1}{1+\exp\left(-v_{i,d}^{t}\right)}$. We used this probability to generate the random numbers $r\in \left[0,1\right]$ and compared it with $TF\left(v_{i,d}^{t}\right)$. If the value was lower than $TF\left(v_{i,d}^{t}\right)$, then it changed its bit value, otherwise it did not change. After converting the approaches of binary space, we added the letter B to indicate the binary space in the method name.

### 3.8. Crowding Distance

The concept of crowd distance was first proposed by [12] in NSGA-II. It measured the density of the solutions in the space with respect to the objectives. The purpose of using this concept was to add a solutions selection criterion when they were non-dominated. Basically, solutions located in less crowded areas or that had bigger crowd distances were favoured. Incorporating this criterion in PSO provided higher exploration in the solution space. The pseudo-code for determining crowd distance is provided in Algorithm 1.
**Algorithm 1.** The algorithm for calculating crowd distance**Input** Pareto front   // set of non-dominated solutions  Output  CD       // crowd distance **Start** N = size of Pareto front  Sorted Pareto Front = sort(Pareto front) initiate CD of size N-2 with zeros  for each solution in the Sorted Pareto Front    find the distance from the previous objective    find the distance from the next objective           calculate the distance of the subject solution CD(i) as the summation           of both the distance from the previous objective and the next objective           endfor  **End**

### 3.9. Developed DAMP Model

The distance angle multi-objective particle swarm optimisation (DAMP) model developed by this study is explained in this section. A directionally aware MOO differs from traditional MOO by exploring power that aims at probabilistic spreading of the solutions in the space according to their crowding distance and directions. Contrary to a study by [13], which determined combined crowding distance with direction in the search and allocated more priority to direction, DAMP probability performs the search while allocating equal weight or selection probability to both direction and distance. The pseudo-code for DAMP is provided in Algorithm 2.
**Algorithm 2.** The algorithm for distance angle multi-objective particle swarm optimisation (DAMP).Inputs f1,f2, …fm   //set of objectives  gmax       //maximum number of generations  sizeOfSwarm   // size of solutions of swarm  Vmax       //maximum velocity  Vmin       //minimum velocity              W,c1,c2     //interial, coefficient of moving toward best personal and coefficient of moving toward best global  angleRes    //angle resolution  **Output**
 PF       //pareto front  gmax      //maximum number of iterations  **Start** Initialize swarm Evaluate(swarm,f1,f2, …fm)  g = 0 While g < gmax newSwarm=[] eaders =Select(swarm) For each particle until sizeOfSwarm  newParticle=Update Position (particle,leaders,w,c1,c2,Vmin,Vmax) particle=Mutation(particle)  add particle to newSwarm  EndFor Repository=Combine(swarm,newSwarm) Swarm=Select(Repository,angleRes) Evaluate(swarm,f1,f2, …fm)  g++ EndWhile PF=ParetoFront(swarm) **End**

To ensure the progress of the search, the algorithm selected the best solution out of a combined pool of solutions from both the original swarm and the swarm after mobility and mutation. The selection was performed via non-dominated sorting and sorted using non-domination criterion ≮. The solutions were then ranked accordingly, where x,y belonged to the same rank, followed by x≮y and y≮x.

Assuming that the solutions were sorted within k ranks, as shown in Figure 1, solutions ranked in $R_{1}$ were the most optimal solutions and dominated over the subsequent solutions. Solutions in rank $R_{2}$ were the second most optimal and, while they dominated over other ranks, they were dominated by solutions in rank $R_{1}$. Additionally, if $\forall \;x,y\in R_{i},\;\mathrm{then},\;x≮y\;\mathrm{and}\;y≮x$. The objective was to select N solutions from the sorted solutions.

As shown in the pseudocode in Algorithm 3, the selection algorithm began by selecting solutions from the most optimal ranks $R_{1},R_{2},\ldots R_{i}$, where the following equations applied:(17)$$R_{1}+R_{2}+\ldots R_{i-1}<N,$$
(18)$$R_{1}+R_{2}+\ldots R_{i}>N$$
$N-(N_{1}+N_{2}+\ldots N_{i-1})$ was then selected from $N_{k}\;$ solutions and was consistent with the exploration. The remaining solutions, up to N, were selected in a way that was consistent with exploration. This method considered two criteria—the angle distribution and the crowding distance distribution. It started by sorting the solutions from the highest to the lowest crowding distance and the highest to the lowest angle range. Angle range rank was defined as the number of selected solutions within an angular sector in the solution space. The first and the second sets of sorted solutions were assigned to sorted solutions distance and sorted solutions angle, respectively. It then underwent an iterative process from 1 until $N-(N_{1}+N_{2}+\ldots N_{i-1})$ and generated a random number (r) in each iteration. If the generated number was between 0 and 0.5, a solution was selected from sorted solutions distance. Otherwise, it was selected from sorted solutions angle. This ensured a balance between both angle and crowding distance explorations. The pseudo-code is provided in Algorithm 4. For DAMP, the solutions selection probability combined two criteria, crowding distance and angle range rank. To demonstrate the concept, we assumed that we had six non-dominated particles, as shown in Figure 2a–c, which shows the corresponding angular range ranks and the crowding distance arrays, respectively. Using the probabilistic calculation, a probability of 0.5/2 was assigned to solutions three and four and then to solutions one and two. An overall probability of 1/4 was assigned to all four solutions.
**Algorithm 3.** The algorithm for selecting N solutions out of 2N pool.**Input** original swarm  modified swarm  N        // sizeOfSwarm Output  selectedSolutions **Start** poolSolutions=combine(original swarm,modified swarm ) SortedSolutions=nonDominatedSorting(poolSolutions) for each rank k lk=length(rank) cumulativeLength=0; if(cumulativeLength<N) add rank k to selectedSolutions cumulativeLength=cumulativeLength+lk  else  RemainingSolution=select(poolSolutions,N-cumulativeLength) add to selectedSolutions end **End****Algorithm 4.** The algorithm for selecting RS solutions out of a pool of non-dominated solutions using crowding distance and angle range rank.**Input** poolSolutions     //repository  RS         //remaining to reach the size of swarm foundSolutions selectedSolutions **Output** selectedSolutions  **Start** center = generateCrowdCenter(foundSolutions) sortedSolutionsDistance=sortingDistance(poolSolutions,center) anglesRangesRank=generateAngleRangeRank(foundSolutions) sortedSolutionsAngles=sortingAngle(foundSolutions) for i = 1 until RS r = generateRandom(0,1) if (r < 0.5) add sortedSolutionsDistance(1) to selectedSolutions delete sortedSolutionsDistance(1) else  add sortedSolutionsAngle(1) to selectedSolutions  delete sortedSolutionsAngle(1) end  end  **End**

### 3.10. Big O Notation

We present the complexity analysis of four variants of MP. Assuming that we had for our meta-heuristic searching N particle, M iterations, the particle length d, the number of objectives m, then the complexity of the searching was given as
$$O\left(M\left\{N\left[O\left(mobilityEquation\right)+O\left(Evaluation\right)\right]+O\left(sorting\right)\right\}\right)$$
$O\left(Evaluation\right)$ was related to the tested mathematical function
$$O\left(mobilityEquation\right)=d$$
$$O\left(sorting\right)=mN^{2}$$
$$O\left(M\left\{N\left[O\left(mobilityEquation\right)+O\left(Evaluation\right)\right]+O\left(sorting\right)\right\}\right)=O\left(MNO\left(Evaluation\right)+MNd+MmN^{2}\right)$$

We observed that the only difference between DAMP, DMP, AMP, and MP was the sorting part where the algorithm had to sort the solutions based on their angle as well as their distance. The sorting part for the three algorithms was the same, which was O ($mN^{2}$), because the big O notation of summation to two functions was the maximum big O notation of them.
(19)$$f_{1}=O\left(g_{1}\right)\;\mathrm{and}\;f_{2}=O\left(g_{2}\right)\Rightarrow f_{1}+f_{2}=O\left(max\left(g_{1},g_{2}\right)\right)$$

### 3.11. Evaluation

The MOO performance measures used to evaluate our proposed method and a comparison between our method and the benchmarking MOO mathematical functions are provided in this sub-section.

#### 3.11.1. C-Metric Measure

C-metric, or set coverage, compared the two Pareto fronts of two approaches in terms of domination. If we had both approach A and approach B, the Pareto front generated from approach A was labelled $P_{A},$ and the Pareto front generated from approach B was labelled $P_{B}$. C-metric $C\left(A,B\right)=C_{B}$ indicated the number of solutions from B that were dominated by solutions in A. The lower the $C_{B}$ value was, the better the performance was. Therefore, the objective was to develop an approach with the lowest $C_{B}$ value. The formula for this measure is expressed in Equation (20):(20)$$C\left(P_{A},P_{B}\right)=\frac{\left|\{y\in P_{B}|\exists_{x}\in P_{A}:x≻y\}\right|}{\left|P_{B}\right|}$$

#### 3.11.2. Hyper-Volume Measure

This measure was a simultaneous indicator of diversity and domination. It was defined as the correlation between the hypercube, the diagonal distance between the solutions in the Pareto front, and the worst set in terms of domination. Therefore, the higher the hyper-volume volume contributed to a better quality of the solutions. The formula for this measure is expressed:(21)$$HV=volume\left(\underset{x\epsilon P_{S}}{\cup }HyperCube\left(x\right)\right)$$

#### 3.11.3. Delta Measure

This measure was a simultaneous indicator of the uniformity of the Pareto fronts’ distributions and spread. Therefore, it was a measure of diversity. It was denoted by Δ and needed to be minimal.
(22)$$\Delta =\frac{d_{f}+d_{l}+\sum_{i=1}^{N-1}|d_{i}-\bar{d|}}{d_{f}+d_{l}+\left(N-1\right)\bar{d}}$$
where:$d_{f}$ and $d_{l}$ were the Euclidean distances between the extreme solutions and the boundary solution,$d_{i}$ was distances where *i* = 1, 2,…, N − 1,$\bar{d}$ was the average of all the consecutive distances $d_{i}$ for *i* = 1, 2,…, N − 1.

The pseudocode for calculating this measure is provided in Algorithm 5, [12].
**Algorithm 5.** The algorithm for calculating the delta measure**Input** d        //Pareto front  dT       //True Pareto Front  Output:  Δ        //Delta Measure  **Start**
 Sort the Pareto set  Calculate the Euclidean distance between consecutive solution and assign them to matrix M Calculate the average of matrix M Fit the curve of the true Pareto front and calculate the distance between the two extreme solutions calculate the distance between the two extreme solutions  Apply equation and find Δ **End**

#### 3.11.4. Generational Distance (GD) 

This measure was an indicator of the optimality of the solutions in terms of their closeness to the true Pareto front solutions. It measured the average distances between the Pareto front solutions and the true Pareto front solutions. Therefore, the lower the GD value was, the more optimal a solution there was [44].
(23)$$GD\left(P_{S},P_{T}\right)=\frac{{\left(\sum_{i=1}^{\left|P_{S}\right|}d_{i}^{2}\right)}^{\frac{1}{2}}}{\left|P_{S}\right|}$$
where:

$\left|P_{S}\right|$ was the number of solutions in the Pareto set,

$P_{T}$ was the true Pareto front,

$d_{i}$ was the Euclidean distance between the solutions in $P_{S}$ and the nearest solutions in $P_{T}$.

#### 3.11.5. Number of Non-Dominated Solutions

This measure was an indicator of MOO algorithm performance in terms of the number of non-dominated solutions (NDS) in the Pareto front. Therefore, the higher the NDS values were, the higher the performance was [45].
(24)$$NDS\;\left(N\right)=\;\left|P_{S}\right|$$

#### 3.11.6. Mathematical Benchmarking Functions

Schaffer (SCH), Fonseca-Fleming (FON), Poloni (POL), Kursawe (KUR), and Zitzler–Deb–Thiele (ZDT1, ZDT2, ZDT3, ZDT4, and ZDT6) functions were used [12]. The mathematical formulas of these functions with optimal solutions and the type of the functions are provided in Table 3.

## 4. Evaluation and Results

DAMP, DMP, and AMP were evaluated using MATLAB 2019b and other benchmark MOPSOs [9], as shown in Table 4. We used the same common parameters of the proposed methods and the benchmarks, and we used the same objective for comparison. Furthermore, each experiment was repeated for 10 runs, changing the seed of the random number generator. The parameters that were selected were based on tuning processes for c1 and c2 that represent the coefficients of the effect of personal best and global best respectively. They were selected to be 1/3 and 2/3, respectively. We also set the parameters of w,$\;V_{max}$, and $V_{min}$ to be 0.5, 0.1, and 0.001, respectively. These parameters were related to the original equation of particles mobility of PSO that are presented in Equations (25)–(28).
(25)$$V_{id}\left(t+1\right)=\omega V_{id}\left(t\right)+C_{1}r_{1d}\left(P_{i}-X_{id}\right)+C_{2}r_{2d}\left(P_{gd}-X_{id}\right),$$
(26)$$X_{id}\left(t+1\right)=X_{id}\left(t\right)+V_{id}\left(t+1\right)$$
(27)$$\mathrm{If}\;V_{id}>V_{max},\mathrm{then}\;V_{id}=V_{max}$$
(28)$$\mathrm{If}\;V_{id}>V_{max},\mathrm{then}\;V_{id}=V_{max}$$

### 4.1. Evaluation By Mathematical Functions

This section evaluates MOO with the MOO mathematical functions mentioned in the sub-section above. The results are presented in the following Section 4.1.1 Set Coverage Analysis, Section 4.1.2 Hyper-volume, Section 4.1.3 Number of Non-Dominated Solutions (NDS), Section 4.1.4 Delta Measure, Section 4.1.5 Generational Distance, and Section 4.1.6 Statistical Evaluation.

#### 4.1.1. Set Coverage Analysis

The role of set coverage was to judge the superiority of the developed DAMP model over industry benchmarks, such as MP and DMP, and intermediate variant models, such as AMP. Therefore, we attempted to compare $C\left(DAMP,x\right)$ values.

With $C\left(x,DAMP\right),$
x could represent any one of the three compared models. Figure 3 shows the obvious superiority of the DAMP model over the C-metric values of all the other mathematical functions. However, a difference existed in domination performance between the models. For instance, the DAMP model had the most domination over the MP model compared to the other two models. We also observed similar domination performance between AMP and DMP models. We also noticed that DAMP had higher $C\left(DAMP,x\right)$ values when evaluated using ZDT1, ZDT2, ZDT3, ZDT4, ZDT6, and FON. On the other hand, we found that the $C\left(DAMP,x\right)$ values were nearly identical when evaluated using POL, and that the $C\left(x,DAMP\right)$ values were slightly lower when evaluated using KUR when x=AMP, DMP.

Therefore, the overall performance of the DAMP model was clearly superior in comparison to all other models in terms of domination.

#### 4.1.2. Hyper-Volume

This function measure judged the diversity of the developed solutions. While it was regarded as a secondary measure, after a set coverage, where domination was more important than diversity, judging diversity helped identify the overall performance of the algorithm. As shown in Figure 4, we observed that the MP model provided higher hyper-volume values than the other models when evaluated using ZDT1, ZDT3, ZDT4, and KUR. Furthermore, Table 5 shows the statistical significance of hyper-volume values for ZDT1, ZDT4, ZDT6, SCH, and FON based on *t*-test values. Cross-references of these visualisation results with statistical results are shown in Table 5 where the superiority of the DAMP model in terms of hyper-volume value is statistically shown for FON, SCH, and ZDT6. Conversely, the hyper-volume value of the MP model was superior to the DAMP when evaluated using ZDT1 and ZDT4. Nevertheless, for all these functions, the MP model had relatively low domination, as seen in the set coverage analysis sub-section. We also found that the DAMP model had higher hyper-volume values than the MP model when evaluated using SCH. We observed that the hyper-volume performance, when evaluated using KUR, was similar to the MP model and superior to the AMP and the DMP models. These findings support the quality of diversity of the DAMP model solutions apart from the observed domination in the previous sub-section.

#### 4.1.3. Number of Non-Dominated Solutions (NDS)

This measure indicated the number of non-dominated solutions which was an indicator of the choices provided to the decision maker after optimisation. As seen in Figure 5, the NDS of the DAMP model was comparable with the AMP and the DMP models when evaluated using SCH, ZDT1, ZDT3, ZDT4, KUR, and POL. Moreover, the NDS of all the other models was superior and statistically significant to the MP model when it was evaluated using ZDT1, ZDT2, ZDT3, ZDT4, ZDT6, KUR, and POL, as shown in Table 5.

#### 4.1.4. Delta Measure

This measure was regarded as a measure of diversity. An observation of the values revealed that the DAMP model had an average diversity compared to AMP and DMP models when it was evaluated using almost all of the functions, whereas the MP model had higher diversity when evaluated using KUR and less diversity when evaluated using FON. This measure was, again, a secondary measure after the domination of solutions. Cross-referencing of the average delta values is shown in Figure 6, and the statistical findings in Table 5 confirmed that the DAMP model was statistically superior to the MP model when evaluated using FON, whereas the MP model was not superior, with a statistical significance only for KUR.

#### 4.1.5. Generational Distance

This measure was an indicator of the closeness between the discovered solutions and the true Pareto solutions. It did not provide accurate description of domination as set coverage. However, it did indicate the distance between the discovered solutions and the true Pareto front. In Figure 7, we could see that, for all mathematical functions, DAMP, AMP, and DMP models had similar performance values and were better than the MP model in terms of distance to the true Pareto front. Cross-referencing of these visualisation results with statistical test values are shown in Table 5, and it shows that the DAMP model was statistically significant over the MP model when evaluated using ZDT3, ZDT4, ZDT6, FON, and KUR. Bear in mind that the results of this measure do not reflect domination performance and should, therefore, be read in conjunction with the set coverage results.

#### 4.1.6. Statistical Evaluation

For evaluation purposes, each measure was generated via 10 experiments using random seed for our DAMP model, the AMP model, and the two industry standard DMP and MP models. As seen in Table 5, our DAMP model dominated almost all MOO functions in comparison to the DMP and the MP models. Furthermore, statistical significance also proved the superiority of our DAMP model over the AMP model for some measures, namely, NDS and GD, when evaluated using ZDTs.

### 4.2. Evaluation Based on Load Balancing Model

This section provides the evaluation of the load balancing in terms of the MOO evaluation metrics. The evaluation was based on number of tasks equal to six and number of nodes in the network equal to 30 nodes. The network is depicted in Figure 8 with the assigned tasks to each of the nodes. This shows that each of the algorithm assigned different tasks to the different nodes where there were different solutions with different performance metrics of the same problem. We also observed that the links of the networks were established to produce connected graphs, which enabled the data exchange while executing the tasks. The parameters and the setting of the evaluation are presented in Table 6.

We added a comparison with two additional algorithms, namely, binary non-dominated sorting genetic algorithm (BN2) and binary particle swarm optimisation (BPSO). We used the same number of solutions and iterations, 50 and 100, respectively. As for objective comparison with the benchmarks, we show the evaluation results in the following Section 4.2.1. Set Coverage, Section 4.2.2. Hyper-Volume, Section 4.2.3 Number of Non-Dominated Solutions, Section 4.2.4 Relative Generational Distance, and Section 4.2.5. Statistical Evaluation.

#### 4.2.1. Set Coverage

For objective evaluation, each of the methods was executed 10 times, and its generated set coverage results were compared with the other methods. The results are provided as boxplots depicted in Figure 9, Figure 10 and Figure 11. As it is observed in Figure 9, BAMP achieved higher set coverage for most approaches, which provided the superiority of using the angle as criterion of searching compared with the other approaches. Another observation was the use of the angle, which provided a wide range of possibilities according to the seed compared with not using the angle or the distance, as it has been shown in C(BAMP, BMP) when compared with C(BMP, BAMP) where the latter was narrower than the former. In addition, we observed that BAMP dominated BN2 with higher percentage than the domination of BN2 over BAMP. Contrary to the angle, the usage of distance as criterion for exploration had less influence on the domination, as it is shown in Figure 10, where the majority of approaches accomplished more dominance than standalone usage of distance represented by BDMP. Another observation was that the usage of angle and distance was better than using the distance solely. This is observed in Figure 11 because the value of C(BDAMP, BDMP) was higher than the value of C(BDMP, BDAMP). However, BPSO generated a non-dominated solution compared with the other benchmark. This solution dominated from a single objective over other objectives, namely, the objective of energy distribution, while it was dominated with respect to other objectives.

#### 4.2.2. Hyper-Volume

In order to evaluate the exploration in the objective space, we generated the hyper-volume. The results of the hyper-volume as boxplots are given in Figure 12. Observing the Figure 12, we noticed that each of the experiments provided different values of hyper-volume. In some experiments, BAMP indicated higher value of hyper-volume (HV), while in others, there was superiority of BDMP. This revealed that the performance of exploration was highly sensitive to the initial seed. However, BN2 outperformed other approaches of swarm family in the hyper-volume.

#### 4.2.3. Number of Non-Dominated Solutions

The other metric that was used to evaluate the multi-objective optimization approaches from the perspective of load balancing was the number of non-dominated solutions, which is given in Figure 13. We observed that the approaches had almost the same NDS except for BMP due to the high non-domination reached by searching within each algorithm.

#### 4.2.4. Relative Generational Distance

The last metric was the relative generational distance, which needed to be minimized. Figure 14 shows that BAMP and BPSO were the best in terms of minimizing this metric compared with the other benchmark. However, it is important to distinguish between lower Euclidean distance and more domination, which is not always associated.

#### 4.2.5. Statistical Evaluation

For thorough evaluation, we conducted a *t*-test to verify the superiority from a statistical perspective. The *t*-test evaluation was conducted based on three metrics, namely, RGD, HV, and NDS. The *t*-test was based on a series of 10 runs, changing the random seed. The evaluation used a confidence level of 0.05 for rejecting the null hypothesis and accepting the statistical significance of the difference between the approaches. As we observe in the Figure 15 and Figure 16, respectively, RGD and HV were the two metrics showing most statistical differences between the approaches, while for NDS, statistical difference was only observed for the comparison with BPSO because it was a single objective optimization and was weak for providing the number of non-dominated solutions, as shown in Figure 17.

## 5. Conclusions and Future Considerations

The study successfully optimised load balancing in software defined networking (SDN) using multi-objective optimisation (MOO) based on particle swarm optimisation (PSO). The industry-benchmark-MP model was expanded to include two additional search criteria, crowding distance and crowding angle. The former provided AMP, the latter provided DMP algorithm, and joining them provided DAMP. In addition, Sigmoid transfer function was then incorporated to convert them to binary, which provided BMP, BAMP, BDMP, and BADMP. The evaluation was decomposed into two phases; the first one was conducted based on benchmarking mathematical functions while the second one was conducted on a developed load balancing SDN model with four objectives: energy (E), energy distribution (D), makespan (T), and renting cost (R). It was found from the evaluation that both AMP and DAMP were superior over DMP and MP in terms of the optimization of the benchmarking mathematical functions. Both BAMP and BDAMP were also superior over BMP and BDMP in terms of the load balancing metrics. Hence, the hypothesis of the superiority of directionality or angle in the optimization was confirmed. Furthermore, it was concluded that using conversion to binary space did not affect the performance of the optimization.

As potential applications for our method, we give edge networks where the users have some applications that need tasks to be executed in real time or with less latency, which requires renting some local nodes for this purpose instead of sending them to the cloud. Other potential applications are for the developed multi-objective optimization, which can be used for various combinatory problems, such as surgeries planning in hospitals [46] and job-shop planning [47].

Several limitations of the approach that can be addressed are as follows. Firstly, it uses fixed angle resolution for dividing the solution space. This might lead to non-stable performance based on the value that is given to the angle. Future studies should explore extending load balancing by adding other objectives, such as node reliability, and incorporating more search criteria in the optimisation algorithm. Another future work is to enable adaptive angle decomposition of the solution space. Secondly, it uses probabilistic selection of non-dominated solutions based on angle or distance in an equal way. Another future work is to make the selection based on adaptive probability of selecting non-dominated solutions. Thirdly, it uses global learning based on moving the particle toward its global best following the conventional mobility equation of PSO. The global learning might lead to premature convergence; a better approach is to use comprehensive learning [48]. Fourthly, we will extend the model to handle dynamical aspects such as running tasks twice on the same node and the cache effect.

## Figures and Tables

**Figure 1 sensors-21-03356-f001:**
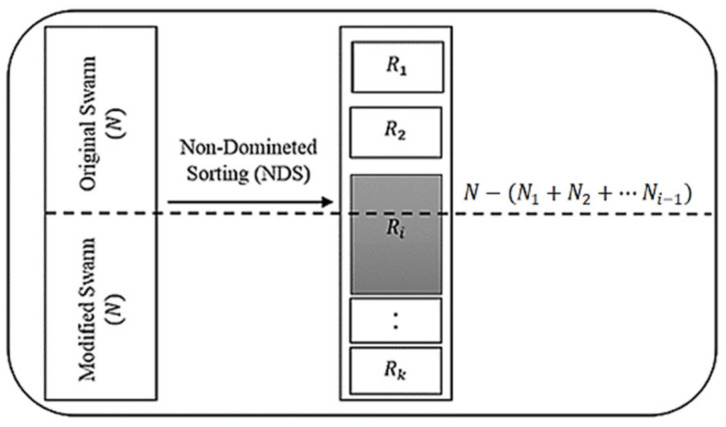
Conceptual diagram for selecting best N solution from 2N pool.

**Figure 2 sensors-21-03356-f002:**
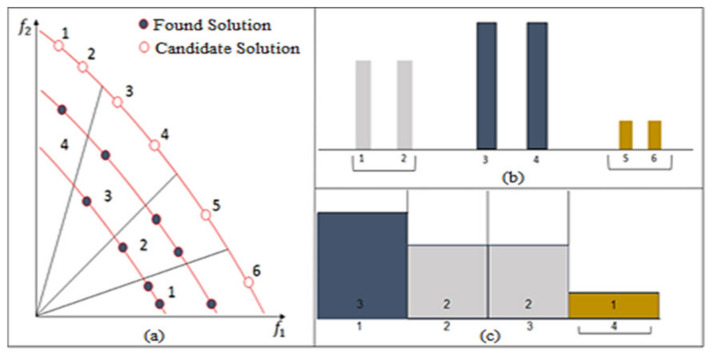
Illustration of candidate solutions selection using DAMP according to the following criteria: (**a**) objective space, (**b**) crowding distance curve, and (**c**) angle range rank curve.

**Figure 3 sensors-21-03356-f003:**
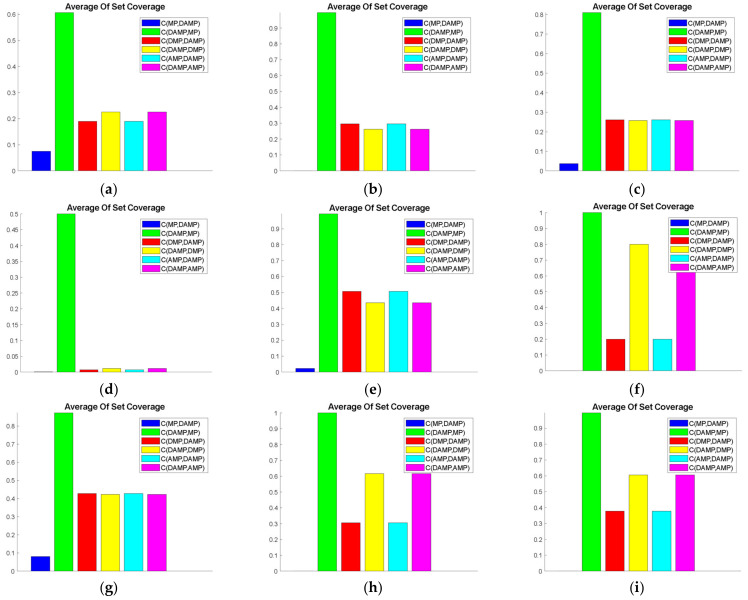
Average set coverage values of the DAMP model compared to MP, AMP, and DMP models: (**a**) FON; (**b**) KUR; (**c**) POL; (**d**) SCH; (**e**) ZDT1; (**f**) ZDT2; (**g**) ZDT3; (**h**) ZDT4; (**i**) ZDT6.

**Figure 4 sensors-21-03356-f004:**
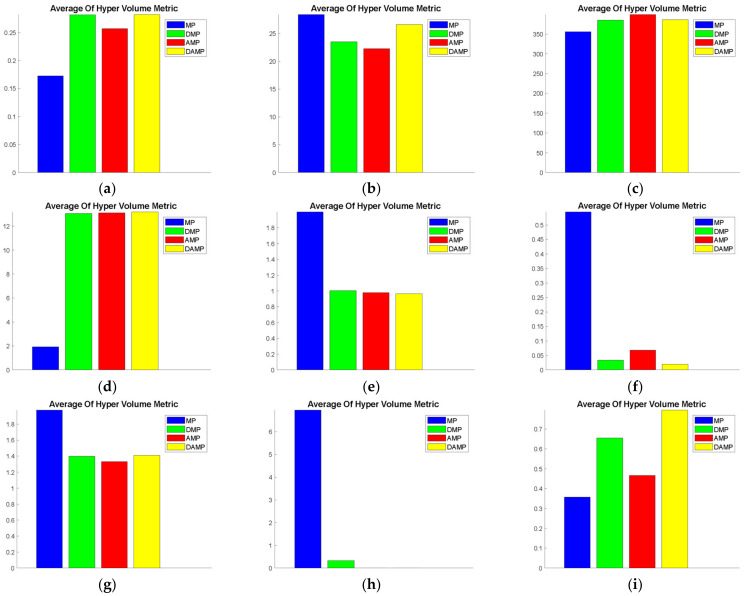
Average hyper-volume values of the DAMP model compared to MP, AMP, and DMP models: (**a**) FON; (**b**) KUR; (**c**) POL; (**d**) SCH; (**e**) ZDT1; (**f**) ZDT2; (**g**) ZDT3; (**h**) ZDT4; (**i**) ZDT6.

**Figure 5 sensors-21-03356-f005:**
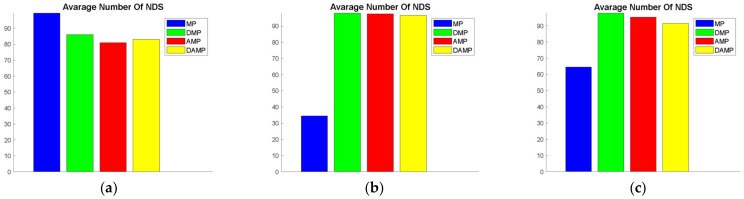
Average non-dominated solutions (NDS) values of the DAMP model compared to MP, AMP, and DMP models: (**a**) FON; (**b**) KUR; (**c**) POL; (**d**) SCH; (**e**) ZDT1; (**f**) ZDT2; (**g**) ZDT3; (**h**) ZDT4; (**i**) ZDT6.

**Figure 6 sensors-21-03356-f006:**
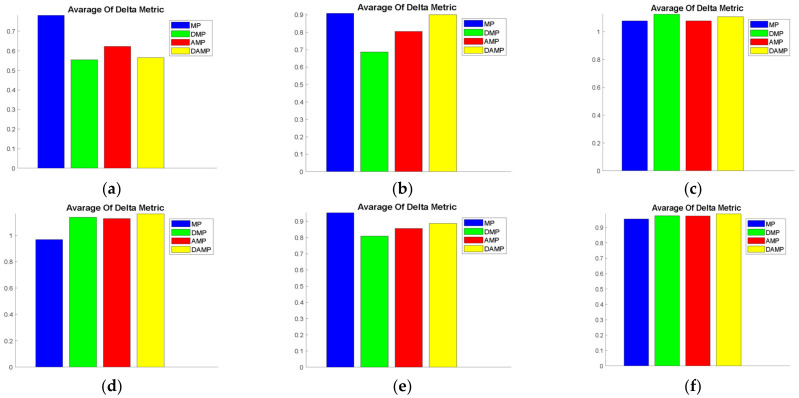
Average delta measure values of the DAMP model compared to MP, AMP, and DMP models: **(a**) FON; (**b**) KUR; (**c**) POL; (**d**) SCH; (**e**) ZDT1; (**f**) ZDT2; (**g**) ZDT3; (**h**) ZDT4; (**i**) ZDT6.

**Figure 7 sensors-21-03356-f007:**
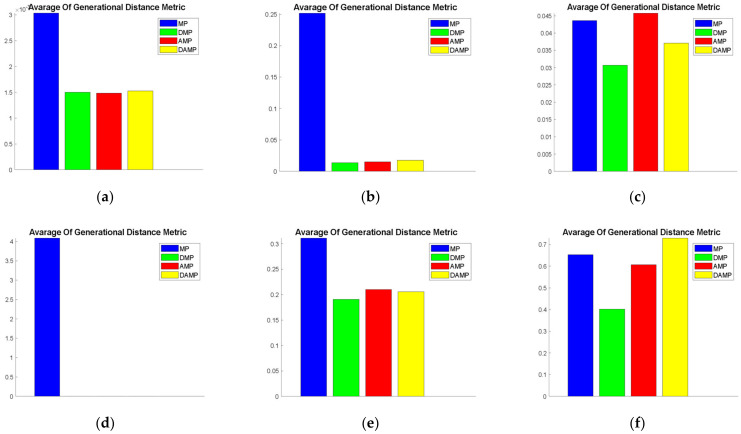
Average MOO function values for generational distance of the DAMP model compared to MP, AMP, and DMP models: (**a**) FON; (**b**) KUR; (**c**) POL; (**d**) SCH; (**e**) ZDT1; (**f**) ZDT2; (**g**) ZDT3; (**h**) ZDT4; (**i**) ZDT6.

**Figure 8 sensors-21-03356-f008:**
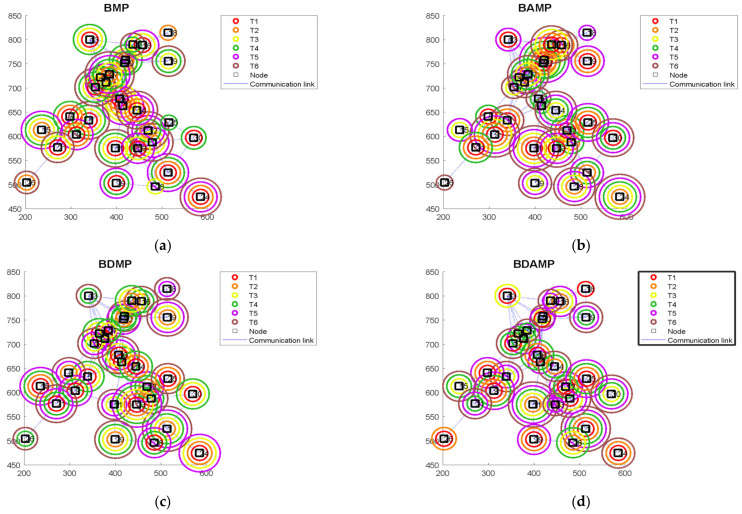
The network used to evaluate the task allocation model: (**a**) binary MP (BMP); (**b**) BAMP; (**c**) BDMP; (**d)** BDAMP.

**Figure 9 sensors-21-03356-f009:**
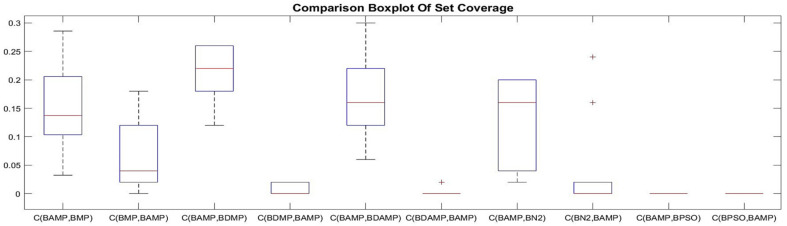
Box plot of the set coverage between the approaches of BAMP with respect to BMP, BDMP, BDAMP, BN2, and BPSO.

**Figure 10 sensors-21-03356-f010:**
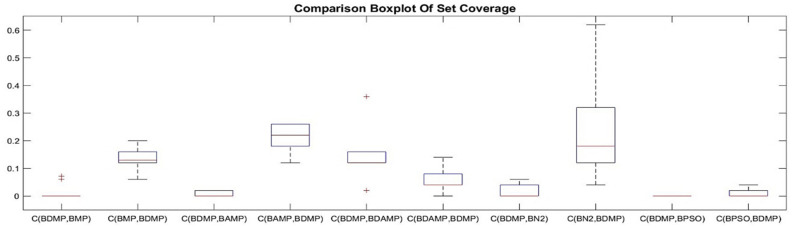
Boxplot of the set coverage between the approaches of BDMP with respect to BMP, BAMP, BDAMP, BN2, and BPSO.

**Figure 11 sensors-21-03356-f011:**
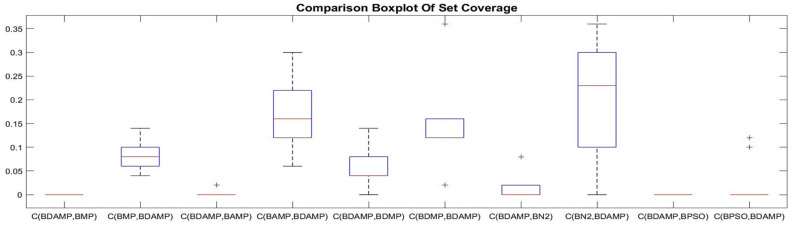
Box plot of the set coverage between the approaches of BDAMP with respect to BMP, BAMP, BDMP, BN2, and BPSO.

**Figure 12 sensors-21-03356-f012:**
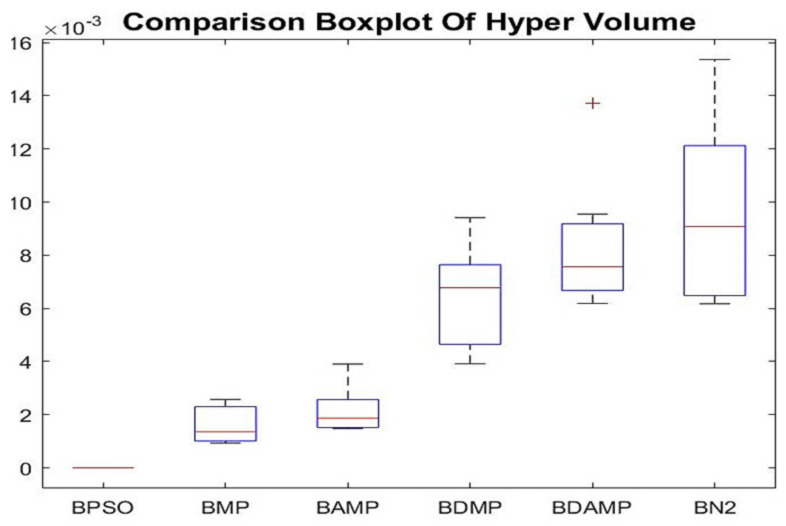
Boxplot of hyper volume results of BAMP and BDAMP compared with benchmarks.

**Figure 13 sensors-21-03356-f013:**
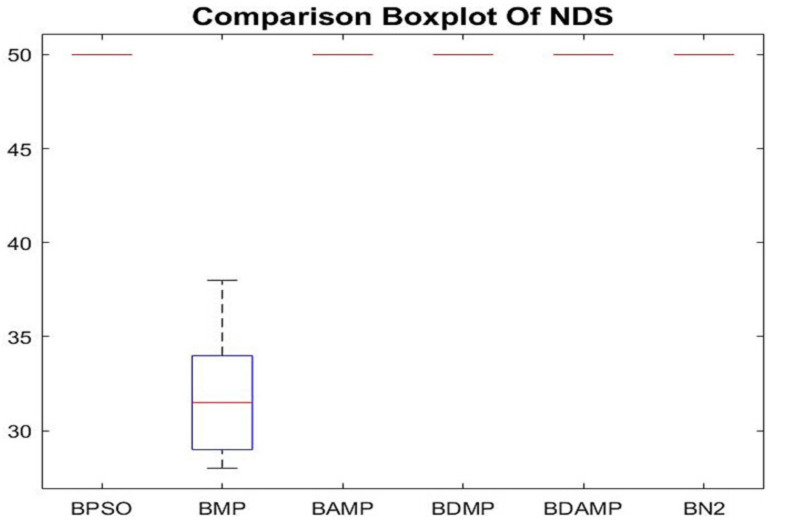
Boxplot of NDS results of BAMP and BDAMP compared with the benchmarks.

**Figure 14 sensors-21-03356-f014:**
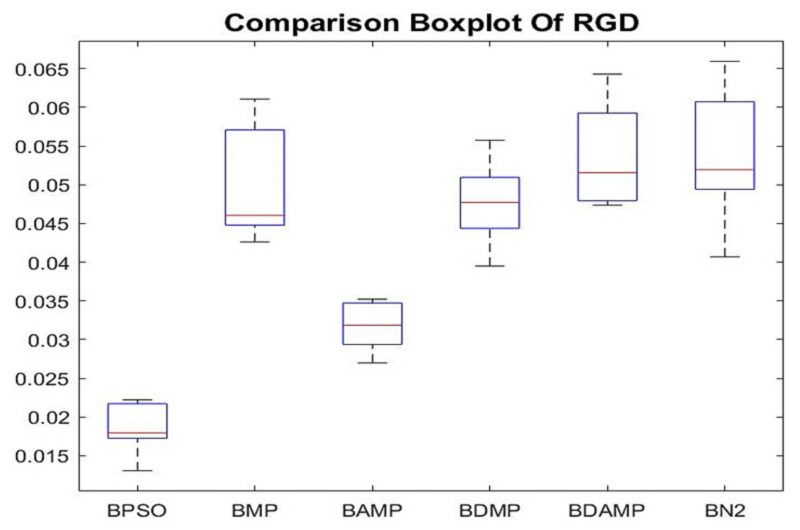
Boxplot of relative generational distance (RDG) results of BAMP and BDAMP compared with the benchmarks.

**Figure 15 sensors-21-03356-f015:**
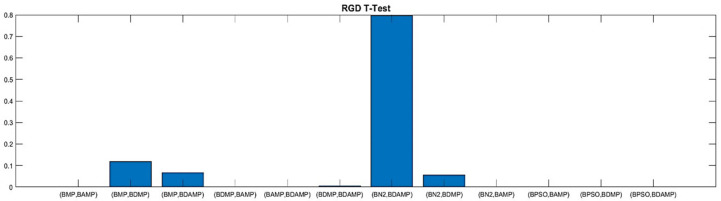
RGD T-test values between our approach and the benchmarks.

**Figure 16 sensors-21-03356-f016:**
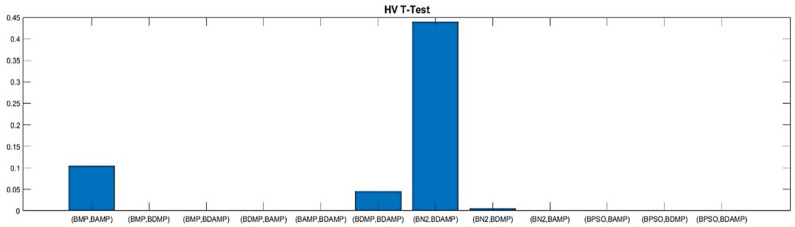
HV T-test values between our approach and the benchmarks.

**Figure 17 sensors-21-03356-f017:**
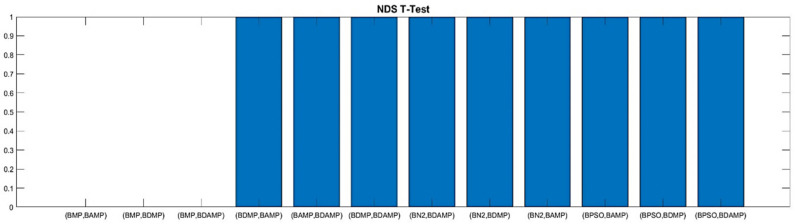
NDS T-test values between our approach and the benchmarks.

**Table 1 sensors-21-03356-t001:** Summary of various existing models and objectives for load balancing.

Authors	Meta-Heuristic	Application	Network Load	Reliability	Energy	Cost	Execution Time
[16]	Discrete Particle swarm optimization Algorithm	Virtual Network Embedding Algorithm for SDN	Yes	Yes	No	No	No
[17]	Chaotic Slap Optimization	Distributed Multi-Controller Deployment	No	Yes	No	No	Yes
[18]	The Reference Vector Based Algorithm	SDN Based Resource Selection	No	No	Yes	Yes	No
[21]	Genetic-Ant Colony Optimization	Traffic Load Balancing.	Yes	No	Yes	No	No
[22]	Genetic Optimization	Load Balancing	Yes	No	Yes	No	No
[23]	Bat Algorithm	SDN Based Load Balancing	Yes	No	No	No	No
[24]	Multi-Objective Particle Swarm Optimization Algorithm	SDN Based Load Balancing	Yes	No	Yes	No	No
Ours	DAMP/AMP	SDN Based Load Balancing	Yes	Yes	Yes	Yes	Yes

SDN: software-defined networking; PSO: particle swarm optimisation.

**Table 2 sensors-21-03356-t002:** Explanation of symbols used throughout this study.

Symbol	Meaning
N	Set of nodes of network
R	Set of connections or edges between nodes
$e_{ij}$	Connection between node i and node j
$N_{j}$	Node of index j
$d_{ij}$	Distance between node i and node j
$v_{j}$	Speed of node j
$e_{j}$	Average energy consumption
$e_{init}$	Initial energy of node
$P_{0}$	Maximum computational load that can be handled by one node
$L_{0}$	Maximum communication load that can be handled by one node
$P_{i}$	Computational load of task i
$L_{i}$	Communication load of task i
K	The number of nodes required to execute the task
M	Set of tasks
$E_{M}$	Set of directions of tasks dependency
$E_{comp}$	Computation energy
$E_{Tx}\left(k,d\right)$	Energy consumption for transmitting k bits for distance d
$E_{Rx}\left(k\right)$	Energy consumption for receiving k bits
$E_{elec}$	Constant to run the transmitter or receiver circuit
$\epsilon_{amp}$	coefficient related to the transmitter amplifier
d	Distance between transmitter and receiver
w	inertia
$V_{max}$	Maximum velocity of particle
$V_{min}$	Minimum velocity of particle
C1	Constant of local target or leader
C2	Constant of global target or leader
RS	remaining solutions to be selected from the swarm

**Table 3 sensors-21-03356-t003:** Benchmark multi-objective optimisation (MOO) functions.

Problem	N	Variables Bounds	Objectives functions	Optimal Solutions	Comments
FON	3	[−4, 4]	$f_{1}\left(x\right)=1-exp\left(-\sum_{i=1}^{3}{\left(x_{i}-\frac{1}{\sqrt{3}}\right)}^{2}\right)$ $f_{2}\left(x\right)=1-exp\left(-\sum_{i=1}^{3}{\left(x_{i}+\frac{1}{\sqrt{3}}\right)}^{2}\right)$	x1=x2=x3	Non-convex
KUR	3	[−5, 5]	$f_{1}\left(x\right)=\sum_{i=1}^{n-1}\left(-10\exp\left(-0.2\sqrt{x_{i}^{2}+x_{i+1}^{2})}\right)\right)$ $f_{2}\left(x\right)=\sum_{i=1}^{n}(|x_{i}{|}^{0.8}+5\sin x_{i}^{3})$	Refer to [12]	Non-convex
POL	2	[−π.π]	$f_{1}\left(x\right)=[1+{\left(A_{1}-B_{1}\right)}^{2}+\left(A_{2}-B_{2}{)}^{2}\right]$ $f_{2}\left(x\right)=[{\left(x_{1}+3\right)}^{2}+\left(x_{2}+1{)}^{2}\right]$ $A_{1}=0.5\sin1-2\cos1+\sin2-1.5\cos2$ $A_{2}=1.5\sin1-\cos1+2\sin2-5.0\cos2$ $B_{1}=0.5\sin x_{1}-2\cos x_{1}+\sin x_{2}-1.5\cos x_{2}$ $B_{2}=1.5\sin x_{1}-\cos x_{1}+2\sin x_{2}-5.0\cos x_{2}$	Refer to [12]	Non-convex, Disconnected
SCH	1	[−$10^{3},10^{3}]$	$f_{1}\left(x\right)=x^{2}$ $f_{2}\left(x\right)={\left(x-2\right)}^{2}$	$x\in \left[0,2\right]$	Convex
ZDT1	30	[0, 1]	$f_{1}\left(x\right)=x_{i}$ $f_{2}\left(x\right)=g\left(x\right)\left[1-\sqrt{\left(\frac{x_{1}}{g\left(x\right)}\right)}\right]$ $g\left(x\right)=1+9\left(\sum_{i=2}^{n}x_{i}\right)\;/\;\;\left(n-1\right)$	$x1\;\in \left[0,\;1\right]$ $x1\;\in \left[0,\;1\right]$ $x1\;\in \left[0,\;1\right]$	Convex
ZDT2	30	[0, 1]	$f_{1}\left(x\right)=x_{1}$ $f_{2}\left(x\right)=g\left(x\right)\left[1-{\left(x_{1}\;/\;g_{\left(x\right)}\right)}^{2}\right]$ $g\left(x\right)=1+9\left(\sum_{i=2}^{n}x_{i}\right)\;/\;\;\left(n-1\right)$	$x1\;\in \left[0,\;1\right]$ $x_{i}=0$ i=2, 3,…,n	Non-convex
ZDT3	30	[0, 1]	$f_{1}\left(x\right)=x_{1}$ $f_{2}\left(x\right)=g\left(x\right)\left[1-\sqrt{\frac{x_{1}}{g\left(x\right)}}-\frac{x_{1}}{g\left(x\right)}\;\sin\left(10\pi x_{1}\right)\right]$ $g\left(x\right)=1+9\;\left(\sum_{i=2}^{n}x_{i}\right)\;/\;\;\left(n-1\right)$	$x1\;\in \left[0,\;1\right]$ $x_{i}=0$ i=2,3,…,n	Convex, Disconnected
ZDT4	10	$x_{1}\in \left[0,\;1\right]$ $x_{1}\in \left[-5,\;5\right],i$ =2,…,n	$f_{1}\left(x\right)=x_{1}$ $f_{2}\left(x\right)=g\left(x\right)\left[1-\sqrt{\frac{x_{1}}{g\left(x\right)}}\right]$ $g\left(x\right)=1+10\left(n-1\right)+\sum_{i=1}^{n}[x_{i}^{2}-10\cos\left(4\pi x_{i}\right)]$	$x1\;\in \left[0,\;1\right]$ $x_{i}=0$, i=2, 3,…,n	Non-convex
ZDT6	10	[0, 1]	$f_{1}\left(x\right)=1-exp\left(-4_{x1}\right)\sin^{6}\left(6\pi x_{1}\right)$ $f_{2}\left(x\right)=g\left(x\right)\left[1-{\left(\frac{f_{1}\left(x\right)}{g\left(x\right)}\right)}^{2}\right]$ $g\left(x\right)=1+9{\left[\left(\sum_{i=2}^{n}x_{i}\right)\;/\;\;\left(n-1\right)\right]}^{0.25}$	$x1\;\in \left[0,\;1\right]$ $x_{i}=0$, i=2,3,…,n	Convex,Non-uniformly Spaced

SCH: Schaffer; FON: Fonseca-Fleming; POL: Poloni; KUR: Kursawe; ZDT: Zitzler–Deb–Thiele.

**Table 4 sensors-21-03356-t004:** Parameters used in the evaluations and their values.

Parameter Name	Value
numberOfParticles	50
numberOfIterations	100
c1	1/3
c2	2/3
nRep	100
w	0.5
$V_{max}$	0.1
$V_{min}$	0.001

**Table 5 sensors-21-03356-t005:** T-values of the DAMP model compared to MP, AMP, and DMP models.

Function Test	T-Test	Measure			
		Delta	Hyper volume	NDS	GD
FON	DAMP/AMP	0.315850996	0.307418865	0.58492454	0.622265942
	DAMP/DMP	0.719826015	0.995765743	0.49363686	0.652045003
	DAMP/MP	5.0242 × 10^−5^	0.000207451	7.39 × 10^−5^	0.049132833
KU	DAMP/AMP	0.316571234	0.136064963	0.54369604	0.573461561
	DAMP/DMP	0.012042727	0.269537335	0.51557762	0.396190518
	DAMP/MP	0.90710705	0.80889372	7.7633 × 10^−7^	0.002574095
POL	DAMP/AMP	0.164387112	0.209533066	0.07679627	0.302064115
	DAMP/DMP	0.351181697	0.871518999	2.89 × 10^−2^	0.393267196
	DAMP/MP	0.351181697	0.871518999	0.0206565	0.593382272
SCH	DAMP/AMP	0.304664635	0.383026813	0.60095345	0.533349004
	DAMP/DMP	0.480290477	0.06290748	0.351256638	0.085563604
	DAMP/MP	0.001485655	1.4044 × 10^−5^	0.05283945	0.142239574
ZDT1	DAMP/AMP	0.142123082	0.904186015	0.05720644	0.636188124
	DAMP/DMP	0.001343439	0.740246281	0.26176837	0.209288404
	DAMP/MP	0.016179938	0.000552411	0.26176837	0.209288404
ZDT2	DAMP/AMP	0.096184093	0.123852474	0.19186711	0.575465083
	DAMP/DMP	0.350353147	0.52239308	0.0815855	0.1455833
	DAMP/MP	0.041056651	0.52239308	0.616373095	0.691184369
ZDT3	DAMP/AMP	0.728433661	0.751269882	0.14275619	0.872306061
	DAMP/DMP	0.000428487	0.948368027	0.106679999	0.206603271
	DAMP/MP	0.592841876	0.948368027	1.8854 × 10^−6^	3.31535 × 10^−5^
ZDT4	DAMP/AMP	0.939868234	0.906971038	0.92851591	0.178865555
	DAMP/DMP	0.013632197	0.341959448	0.81800295	0.140962795
	DAMP/MP	0.013632197	0.000838889	0.04702054	8.29888 × 10^−7^
ZDT6	DAMP/AMP	0.534051523	0.06054342	0.09434971	0.058047202
	DAMP/DMP	0.224179011	0.749439813	0.051886	0.099316485
	DAMP/MP	0.224179011	0.005584877	0.00128414	1.34771 × 10^−5^

GD: generational distance.

**Table 6 sensors-21-03356-t006:** Parameters and setting of evaluation.

Parameter Name	Value
number of nodes	6
number of tasks	30
transmission range	100
speed	30 until 100 IPS
power consumption	4 until 10 mW
initial energy	2
Eelc	50 × 10^−6^ [mj/b]
epsilonAmp	10 × 10^−9^ mJ/b/m^2^
$P_{0}$	40 MIPS
$L_{0}$	50 Byte

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
