# Peer review of "Directionally-Enhanced Binary Multi-Objective Particle Swarm Optimisation for Load Balancing in Software Defined Networks"

_sensors, 2021, doi:10.3390/s21103356_

Round 1
Reviewer 1 Report
Authors have improved the paper significantly. Many concerns related to the tasks were not addressed:
All the aspects related to the tasks and how they define performance metrics. They have improved it (moving from cycles per second to IPS -instructions per second-) but for example they do not ask how to characterize KCC ... for example in a EC2 node in an AWS instance on EVM.
When I point that doing what they do is not correct to measure performance metrics because if you run an app twice the cache will hold data to boost it. And this improvement comes for free. Author say this is Out of the Focus of the paper and that they tried to respond as much as possible to my concerns.
So as these issues are secondary .... we can hold this debate for several review rounds...
Author Response
Dear reviewer,
Thank you for your valuable comments. Please see the attachment for the response.
Thank you.

Reviewer 2 Report
Multi-objective optimization in Internet of Things (IoTs) networks is a challenging problem. This paper first presented a joint mathematical formulation to solve load balancing challenges in cloud computing, then proposed two multi-objective particle swarm optimization (MP) models to solve this problem. Experiment results show that their DAMP and AMP models were far superior to the industry-standard models, MP and DMP. Particle Swarm Optimization methods were proposed to end the Multi-Objective problems several decades ago. There are many existing classical PSO or PSO based methods. The authors should compare their methods with them in their paper. Moreover, throughout this paper, I can not find what the shortcomings of their algorithms, which should be described in detail.
Some issues should be addressed correctly, which include, but are not limited to:
- Line 22, in Abstract, “angle angle multi-objective”, double ANGLE?
Line 233, is “}” missing in the definition of R? Moreover, since you use N to represent the node set, the elements in this set should be represented using little character.
- I don’t know why some colored parts and some revisions exist in this paper. For example: Line 6 (the affiliation), Line 238-240, Line 247-248…
- In formula (1), are Pi/P0, Li/L0 integers since k0 is an integer? Should round operation be taken here?
- Some characters in this paper are defined several times. For example, k in formula (3) (4)(5) is the number of transmitted bits, but in formula (1), it is the max of three numbers related to the number of nodes. Another k in Line 348 is used to represent the number of solutions. Thus, one k has at least three meanings. Please check it.
- Line 406~407, some key words are missing.
Author Response

(The authors gave the same response as above.)

Round 2
Reviewer 2 Report
All my comments have been addressed well. It can be published in this journal.